# Proton Microbeam Targeted Irradiation of the Gonad Primordium Region Induces Developmental Alterations Associated with Heat Shock Responses and Cuticle Defense in *Caenorhabditis elegans*

**DOI:** 10.3390/biology12111372

**Published:** 2023-10-27

**Authors:** Pierre Beaudier, Guillaume Devès, Laurent Plawinski, Denis Dupuy, Philippe Barberet, Hervé Seznec

**Affiliations:** 1University Bordeaux, CNRS, LP2I, UMR 5797, 33170 Gradignan, France; beaudier@cenbg.in2p3.fr (P.B.); guillaume.deves@cnrs.fr (G.D.); plawinsk@cenbg.in2p3.fr (L.P.); philippe.barberet@u-bordeaux.fr (P.B.); 2University Bordeaux, INSERM, U1212, 33607 Pessac, France

**Keywords:** microbeam, *Caenorhabditis elegans*, gonadal–vulval development, COPAS, nanopore sequencing

## Abstract

**Simple Summary:**

We used the charged-particle microbeam of the AIFIRA facility to investigate the effects of targeted irradiation of the progenitor gonad stem region on the organogenesis of worm gonad and vulva. A dedicated experimental approach was developed to enable the manipulation and targeted irradiation of the progenitor gonad stem region. This was achieved using three MeV protons during a definite developmental stage of the worms. The outcomes revealed distinct developmental modifications and specific gene inductions involved in cellular stress and cuticle injury responses. These findings were proven through an integration of methodologies, encompassing micro-irradiation under the reversible immobilization of worms, confocal imaging, cell sorting assays, and long-read sequencing analysis.

**Abstract:**

We describe a methodology to manipulate *Caenorhabditis elegans* (*C. elegans*) and irradiate the stem progenitor gonad region using three MeV protons at a specific developmental stage (L1). The consequences of the targeted irradiation were first investigated by considering the organogenesis of the vulva and gonad, two well-defined and characterized developmental systems in *C. elegans*. In addition, we adapted high-throughput analysis protocols, using cell-sorting assays (COPAS) and whole transcriptome analysis, to the limited number of worms (>300) imposed by the selective irradiation approach. Here, the presented status report validated protocols to (i) deliver a controlled dose in specific regions of the worms; (ii) immobilize synchronized worm populations (>300); (iii) specifically target dedicated cells; (iv) study the radiation-induced developmental alterations and gene induction involved in cellular stress (heat shock protein) and cuticle injury responses that were found.

## 1. Introduction

The development of a multicellular organism involves the generation of many different cell types from a single fertilized egg cell. Understanding the molecular mechanisms by which cells become differentiated from each other is a fundamental goal of developmental biology. Our goal is to study how exposures to ionizing radiation (IR) alter or modify these cellular mechanisms.

The nematode *Caenorhabditis elegans* (*C. elegans*) stands as an ideal model organism for dissecting cellular intricacies at the single-cell level. The developmental trajectory of *C. elegans*, encompassing cell divisions, migrations, and differentiations, has been meticulously documented [1,2,3,4]. The pursuit of this comprehensive understanding has been aided by three key attributes intrinsic to *C. elegans*. Firstly, its little size (1 mm for adults) and ease of cultivation on nonpathogenic *Escherichia coli* (*E. coli*) in laboratory settings render it prolific, with a mere three-day life cycle yielding a multitude of progeny from a single individual. Secondly, the organism’s transparency facilitates in situ cell observation. Its consistent cellular lineage, meticulously tracked through direct microscopic observation, remains largely invariant. *C. elegans* have a fixed number of only 558 larval and 959 adult somatic cells, which constitute organs such as muscles, intestine, and neurons. The germ line constitutes a substantial portion of the body, featuring germ line stem cells that sequentially produce sperm and oocytes that will undergo fertilization in the spermatheca and advance to early cleavage stages before emergence via the vulva. Lastly, the prevalence of self-fertile hermaphroditism among *C. elegans* greatly facilitates the generation of inbred lines and genetic analyses. Numerous investigations have delved into the spectrum of cell-to-cell interactions using increasingly sophisticated strategies and phenotypic evaluations facilitated by green fluorescent protein (GFP)-tagged strains to facilitate phenotype detection with a diverse range of documented mutants and transgenic strains (a vast collection which is available at the *Caenorhabditis* Genome Center, University of Minnesota).

A strategy for investigating cellular functionality is to eliminate individual cells, subsequently observing the resultant developmental or behavioral abnormalities. This method is commonly executed through the precise killing of specific cells or cell groups using a laser microbeam, a technique known as laser ablation. Notably, laser ablation has been extensively used to uncover the roles of diverse mature cell types, encompassing neurons engaged in locomotion, feeding, mechano-sensation, and chemo-sensation [5,6,7,8,9]. In addition, the elimination of distal tip cells within the somatic gonad triggers premature germ line differentiation, thus revealing that the somatic gonad is essential for sustaining an undifferentiated state within the germ line [4,10,11]. Laser ablation has also proven particularly insightful for comprehending postembryonic cell interactions during gonad and vulva development, yielding substantial insights into these intricate processes [4,10,11,12]. Extending this approach to charged-particle microbeams could help to characterize the specific radiation-induced biological responses. Charged-particle microbeams provide the possibility of selectively irradiating living cells with a precise dose and dose delivery timing [13,14,15,16,17,18]. Initially developed to target adherent mammalian cells, the use of charged-particle microbeams have progressively broadened to encompass the irradiation of three-dimensional tissue models and small in vivo multicellular system [19,20,21,22]. Amongst the well-characterized biological models, the nematode *C. elegans* is particularly suited to the specific constraints of targeted irradiations [16,23,24,25]. And several reports have detailed the use of *C. elegans* at charged-particle microbeam end stations [16,17,23,24,26,27,28].

The *C. elegans* vulva serves as an exemplary model to unravel the intricate mechanisms governing pattern formation and morphogenesis [29]. The cell divisions of vulval development occur over a period of only five hours and can be readily and directly analyzed in living animals using genetic and molecular methods. It has been demonstrated that cell signaling is central to vulval development. The vulva is induced by the gonad from the epithelium that envelops the animal. Inductive signaling seems to function by preventing the action of another intercellular signal that otherwise inhibits vulval development. Moreover, the responding vulval precursor cells interact to help to establish the precise spatial pattern of expression of vulval cell fates. Morphogenesis of the vulva culminates in a process called vulval eversion, whereby a passageway is made from the uterus through the vulva to the outside of the animals. Mutations causing abnormal eversion of the vulva (*evl* mutants) have been identified. Using laser ablation, it has been shown that the uterus and anchor cell are important for correct vulval eversion. It has been also described that *evl* mutations may influence vulval eversion by primarily affecting the development of the somatic gonad, while others affect the developments of both structures [30].

In the present study, we have applied charged-particle microbeams to selectively irradiate the stem progenitor gonad region (identified by the position of the progenitor gonad stem cells named Z2-Z3) in *C. elegans* at a defined developmental stage (synchronized population of L1 larvae). This approach provides a way to investigate the radiation-induced effects in a living organism in development using a well-defined and well-characterized organogenesis system: gonads and vulva. In addition, this work aimed to combine the use of selective and targeted irradiation and the use of a high-throughput nematode fluorescent analysis device (Complex Object Parametric Analysis and Sorting—COPAS, Union Biometrica) and third generation sequencing (Oxford Nanopore technologies) with a limited number of individuals (<300).

## 2. Materials and Methods

### 2.1. Worm Strains and Culture

*C. elegans* worm strains were cultivated on nematode growth medium (NGM) agar plates and provided unrestricted access to *Escherichia coli* strain (*E. coli*) OP50 at 19 °C, following established protocols [31,32]. We used the transgenic GZ264 strain carrying an appropriate fluorescent marker (GZ264 (isIs17[pGZ265:pie-1::GFP-pcn-1(W0D2.4)]), specifically expressed at the L1 stage in worms, onto the Z2-Z3 progenitor gonad stem cells. The *Caenorhabditis* Genetics Center (CGC, University of Minnesota, Minneapolis, MN, USA) provided these *C. elegans* strains and the *E. coli* OP50.

### 2.2. Preparation of Large Population of Synchronized C. elegans (L1 Stage)

The bleaching technique was used for synchronizing *C. elegans* cultures at the first larval stage (L1). Populations of young gravid hermaphrodites from standard, well-fed culture stocks were collected with M9 buffer (3 g·L^−1^ KH_2_PO_4_ (Sigma-Aldrich, Saint-Quentin Fallavier, France, cat. no. P9791-500G), 6 g·L^−1^ Na_2_HPO_4_(Sigma-Aldrich, Saint-Quentin Fallavier, France, cat. no. S5136-500G), 5 g·L^−1^ NaCl (Sigma-Aldrich, Saint-Quentin Fallavier, France, cat. no. S3014-500G), and 1 mM MgSO_4_ (Sigma-Aldrich, Saint-Quentin Fallavier, France, cat. no. M2643-500G)) and washed three times with sterile water to remove bacteria. Then, worms pelleted via centrifugation (2 min, 2000 rpms, room temperature) were treated with a freshly prepared alkaline hypochlorite solution (1.5% (*v*/*v*) NaOCl (Merck KGaA, Darmstadt, Germany, cat. no. K50436114829) and 1 M NaOH (Sigma-Aldrich, Saint-Quentin Fallavier, France, cat. no. S5881-500G-M). The suspension was swirled every 2 min with vortex-mixing (~6 min). The released embryos were pelleted via centrifugation (2000 rpms; 2 min; 4 °C)**.** The supernatant was carefully discarded, and embryos were washed three times with M9 buffer followed by centrifugation. The pelleted embryos were suspended in M9 and plated on an NGM agar plate without bacteria. The elapsed time between hatching and irradiation was reduced in order to favor healthy conditions (<20 h).

### 2.3. Samples Preparation and Mounting for Irradiation

We adapted the sample preparation conditions to use our custom-made support dish for micro-irradiation and live imaging. The day of irradiation, L1 larvae were collected by centrifugation (2000 rpms; 2 min; 4 °C) in the cold M9 medium and resuspended in the mounting medium (M9 supplemented with 0.25 mM Tetramisole hydrochloride (Sigma-Aldrich, Saint-Quentin Fallavier, France, cat. no. L9756-10G), with 30% (*v*/*v*) Poloxamer-407 (Sigma-Aldrich, Saint-Quentin Fallavier, France, cat. no. 16758-500G). L1 were stored at 4 °C to avoid their immobilization by the polymerization of the mounting medium at 20 °C. The number of worms was estimated to adjust the dilution volume to obtain ~100 worms per µL. 30 min before irradiation, an aliquot ~2 µL was directly deposited on a sterile 4 μm thick polypropylene (Goodfellow Cambridge Ltd., Huntingdon, UK, cat. no. PP301040) and immediately covered with an afresh agar pad (3% (*w*/*v*), Sigma-Aldrich, Saint-Quentin Fallavier, France, cat. no. A1296-1KG) in M9 to maintain worm immobilized in a thin layer of medium. To prevent desiccation and contamination, the dish was closed with a glass side coverslip. The elapsed time between mounting and irradiation was limited to 1 h.

### 2.4. Beam Line Characteristics and Irradiation Procedures

Worms were irradiated with 3 MeV protons delivered on the micro-irradiation setup installed on the AIFIRA facility (Applications Interdisciplinaires des Faisceaux d’Ions en Région Aquitaine, Bordeaux, France) [14,15]. The accelerator (Singletron^TM^, High Voltage Engineering Europa, Amersfoort, The Netherlands) delivers light ion beams with energies up to 3 MeV [33]. To target Z2-Z3 cells, the beam was strongly collimated to reduce the particle flux to a few thousand protons per second on the target and was focused using a triplet of magnetic quadrupoles. Before hitting the target, the protons pass through a 200 nm thick Si_3_N_4_ membrane (Silson Ltd., Southam, UK) and a 700 µm thick residual air layer, leading to a beam spot size of about 4 µm. The beam was targeted to Z2-Z3 thanks to the use of the GZ264 transgenic strain. Indeed, at the first larval stage (L1), of the whole organism, only these two cells expressed fluorescence based on GFP-tagged protein position (GFP::PCN-1). The beam spot raster scanned on a 16 × 16 µm^2^ area to ensure a homogenous dose distribution on the gonad stem cells while preserving the worm body from irradiation. Assuming a constant LET of the protons (12 keV·µm^−1^) when traversing the worms, and for a given fluence F, the mean dose was calculated as follows:$$D\left(Gy\right)=1.6\times 10^{-9}\cdot LET\left(keV\cdot µm^{-1}\right)\cdot F\left(cm^{-2}\right)$$

For each replicate, we irradiated a minimum of 3 independent worm populations, consisting of 300 to 450 worms in each population. This adds up to a total of 900 to 1200 irradiated worms per condition for every replicate [33,34]. The irradiation end station consists of a motorized inverted fluorescence microscope (Carl Zeiss Micro-Imaging S.A.S, Rueil-Malmaison, France, cat. no. AxioObserver Z1) equipped with a 14 bits Rolera EM-C^2TM^ Camera (Teledyne Photometrics, Tucson, AZ, USA, cat. no. QImaging) which is positioned horizontally at the end of the beam line. It is equipped with 63× objective (Carl Zeiss Micro-Imaging S.A.S, Rueil-Malmaison, France, cat. no. LD Plan-Neofluar 20×/0.4). Fluorescence light is provided by a light emitting diode (LED) illuminating system (Carl Zeiss Micro-Imaging S.A.S, Rueil-Malmaison, France, cat. no. Colibri2^TM^) with negligible heat production. The image acquisition is performed using the Micromanager software [35]. 

### 2.5. Sample Preparation for Confocal Imaging

Worm populations were immediately fixed in cold 4% (*w*/*v*) paraformaldehyde (Sigma-Aldrich, Saint-Quentin Fallavier, France, cat. no. 158127-500G) for 15 min. Then, worms were pelleted via centrifugation (2 min., 2000 rpms, RT) and paraformaldehyde was replaced by cold acetone (Sigma-Aldrich, Saint-Quentin Fallavier, France, cat. no. 650501-1L) for permeabilization (2 min, −20 °C) and finally washed twice in M9. After fixation and washing, M9 was removed and replaced by a freshly prepared solution of DyLight 594 Phalloidin (1:80 (*v*/*v*), Cell Signaling Technologies, Ozyme, Saint-Cyr-L’École, France, cat. no. 12877) and Hoecsht^33342^ (2:4000 (*v*/*v*), Thermo Fischer Scientific, Illkirch, France, cat. no. H3570). Worms were incubated overnight at RT under gentle agitation and then washed via two series of centrifugation (2 min, 2000 rpms) with M9. The supernatant was discarded, and the pelleted worms were suspended in 2–3 drops of Prolong Gold Antifade reagent (Thermo Fischer Scientific, Illkirch, France, cat. no. P36934) and transferred via pipettes for mounting between the glass slides. Three-dimensional images were acquired with a Leica DMRE TCS SP2 AOBS confocal microscope (oil-immersion objective × 63, 1.4 NA), assembled, and reconstructed using Image J software (Rasband, W.S., ImageJ, U.S. National Institutes of Health, Bethesda, MD, USA, https://imagej.nih.gov/ij/, accessed on 20 September 2023, 1997–2018).

### 2.6. Nematode Population Analysis and Gene Reporter Assay Using COPAS

Proper larval development is essential for the normal function of *C. elegans* in adult life stages. Delayed growth in *C. elegans* can be indicative of dysfunction of a number of developmental processes and can be measured in a high-throughput growth assay using the Complex Object Parameter Analysis Sorter (COPAS Biosort, Union Biometrica, Aalst, Belgium) [36]. The COPAS Biosort is a flow cytometer designed specifically for large object analysis. It records worm axial length (time of flight, TOF), optical density (EXT), and longitudinal fluorescence variation in worms. 

COPAS Biosort calibration and settings were defined analyzing a non-synchronized GZ264 population. A 488 nm laser source is used to excite GFP in worms as its passes through the flow cell. GFP intensity is then recorded using a combination of a dichroic mirror reflecting emitted fluorescence (510/523 nm) and a photomultiplier tube (PMT). PMT scale, gain, and threshold were kept constant during all acquisitions.

After irradiation and recovery time, populations were carefully collected by rinsing the Petri dish with ultrapure water. Suspended worms were then analyzed using the COPAS device. The dilution of worms was adjusted to keep the counting rate in the range of 10–50 per second. No gating of experimental parameters (TOF, EXT, and PhGreen (Peak Height Green)) was applied during acquisition. A dedicated python library was developed for data analysis. For each sample, it provides (i) the number of worms, (ii) their size (body length evaluated by TOF), and (iii) the expression level of the gene reporter per individual, along with its longitudinal profile.

Standard procedure to produce GFP longitudinal intensity profiles using our python library includes the following steps: reading raw data; filtering events, if any, corresponding to measure artefacts like bubbles; sorting individual profiles by length and filtering profiles with a length smaller than 80 in order to discard small objects (debris, eggs, or younger worms); aligning profiles from head to tail using a Pearson’s correlation test between adjacent individuals; and finally, merging worm profiles with the same length, producing mean profiles for each observed length.

### 2.7. RNA Collection

Efficient extraction to obtain a good amount and high quality of RNA remains critical in the success of transcription analysis. As our samples were not immediately processed for RNA extraction, they were collected by centrifugation in M9 buffer (3 h post-irradiation) and stored at −80 °C until analysis. Although *C. elegans* has proved to be a powerful tool for genetic analyses, its use has been stalled by the inability to efficiently and rapidly break its resistant cuticle that must be ruptured prior to RNA extraction. Thus, we adapted the protocol established by Qiagen (RNeasy Kit, Courtaboeuf, France), which allows efficient purification of total RNA from small amounts of starting material. 

We need to consider that our primary goal is to minimize the time required to irradiate all the worms of a sample. Increasing the number of worms per sample presents several challenges. Worm density is a critical factor in our ability to accurately distinguish stem cells within their bodies, as overcrowding can lead to overlapping worms. Moreover, worms are sensitive to high-density populations, which can induce stress. The selected number of worms (>450) is the most suitable for our experimental conditions. Furthermore, even though we irradiated only 450 worms per sample, we irradiated several separate samples (at least 3). Since these samples were irradiated at different times, we need to perform distinct extractions at various intervals after the irradiation process.

After irradiation, L1 worms were cultured for 3 h on NGM plates seeding with an *E. coli* OP50 strain, washed in RNase-free water, and pelleted by centrifugation to remove bacteria. Samples need to be first lysed in RLT buffer and then homogenized and purified using a silica membrane via successive washing and centrifugation steps. We tested two methods to disrupt the cuticle of frozen worms using (i) bead beating with Precellys (Bertin Technologies) or (ii) Dounce homogenization combined with successive and alternate freeze-cracking steps. In the bead beating procedure, microscopic glass beads are used to mechanically disrupt the cuticle. Although this method is cheap and fast, variability in the RNA material recovery can be observed; it also requires a relatively large number of worms, and the RNA tends to degrade as the samples heat up. By contrast, Dounce homogenization combined with successive and alternate freeze-cracking steps resulted in an acceptable RNA quantity and reproducible quality from a small number of L1 populations (<450). In this procedure, worms were lysed via 20 cycles of freeze-cracking using a Dounce tissue homogenizer (Sigma-Aldrich, Saint-Quentin Fallavier, France, cat. no. 885301/885303) and total RNA was isolated with an RNeasy Mini kit according to manufacturer’s instructions (Qiagen SAS, Courtaboeuf, France, cat. no. 74104). Total RNA integrity was assessed with the Agilent high-sensitivity RNA system for TapeStation (Agilent Technologies, Les Ulis, France).

### 2.8. Library Preparation and Sequencing (Oxford Nanopore Technologies, PCR-cDNA Barcoding)

Whole transcriptome cDNA libraries were first constructed from extracted mRNA using a PCR-cDNA barcoding kit (Oxford Nanopore Technologies, Oxford, UK, cat. no. SQK-PCB109). Briefly, 50 ng total RNA (~1 ng (polyA)+ mRNA) from each condition was taken in RNase-free PCR tubes containing Maxima H Minus reverse transcriptase enzyme (Thermo Fischer Scientific, Illkirch, France, cat. no. EP0751), VN primer (2 µM, variant of oligo (dT) with complementary nucleotides for annealing of barcode primers), 10 mM dNTPs (New England Biolabs, Evry, France cat. no. N0447S), RNaseOUT (40 U/µL, Thermo Fischer Scientific, Illkirch, France, cat. no. 10777019), and strand-switching primer (10 µM, Oxford Nanopore Technologies Oxford, UK). Products were sequentially added and prepared as recommended by the manufacturers. Samples were incubated at +42 °C for 90 min, followed by enzyme inactivation at +85 °C for 5 min (one cycle) in a thermocycler for the generation of full-length cDNA from poly(A)+ messenger RNA. cDNA libraries from each sample then underwent full-length amplification and sample barcoding using 14 cycles of PCR. Each PCR reaction mix consisted of 25 µL 2× LongAmp Taq master mix (New England Biolabs, Evry, France, cat. no. M0287S), 1.5 µL barcoded primers (named BP01 to BP12), 18.5 µL nuclease-free water, and 5 µL (~0.25 ng) cDNA. The following PCR cycling conditions were used: initial denaturation at 95 °C for 30 s (1 cycle), denaturation at 95 °C for 15 s (14 cycles), annealing at 62 °C for 15 s (14 cycles), extension at 65 °C for 15 s (14 cycles), and final extension at 65 °C for 6 min (1 cycle). (The long extension time of 6 min was to selectively amplify cDNAs up to ~5 kb in length.) 1 µL Exonuclease 1 (New England Biolabs, Evry, France, cat. no. M0293S) was finally added in each PCR tube for incubation at +37 °C, followed by enzyme inactivation at +80 °C for 5 min (one cycle). To clean up the cDNA libraries, PCR reactions with the same barcode (from BP01 to BP12) were pooled in two 1.5 mL DNA tubes and primer dimers were removed using 0.8× volume equivalent Agencourt^®^ AMPure^®^ XP beads (Beckman Coulter, Villepinte, France, cat. no. A63880). Briefly, the beads (80 µL) were added to each pooled sample, incubated on a hula mixer for 5 min at room temperature, and spun and pelleted on a magnet. The supernatants were pipetted off and the resulting beads were washed with 70% (*v*/*v*) ethanol (200 µL, freshly prepared using nuclease-free water) without disturbing the pellet. The ethanol was removed using a pipette and the beads were washed again with ethanol; the pelleted beads were spun down and placed back on the magnet. Residual ethanol was pipetted off and the beads were briefly allowed to dry. While the beads still appeared glossy (with no cracking), they were resuspended in 12 µL elution buffer (provided with the PCR-cDNA barcoding kit; ONT) to recover the cDNA libraries. Prior to loading the barcoded cDNA libraries onto the flow cell for long-read sequencing, 100 ng of each cleaned up cDNA library was ligated with adapters. The barcoded libraries (100 fmol final) were pooled in 1.5 mL DNA, combined with rapid adapters (Oxford Nanopore Technologies, Oxford, UK, cat. no. SQK-PCB109), and incubated at room temperature for 5 min. The adapter-ligated pooled cDNA library was transferred to a fresh tube and combined with loading beads and sequencing buffer (SQK-PCB109 kit; ONT) to form the sequencing mix. The flow cell R9.4.1 (Oxford Nanopore Technologies, Oxford, UK, cat. no. FLO-MIN106D) was primed with pre-mixed flush buffer and flush tether (Oxford Nanopore Technologies, Oxford, UK, cat. no. Flow Cell Priming kit EXP-FLP002), then loaded with the sequencing mix and run for 24 h on a MinION Mk1C sequencing device (Oxford Nanopore Technologies, Oxford, UK).

### 2.9. Transcriptome Analysis

Upon completion, guppy 6.1.4 was used to perform super high accuracy base-calling and data demultiplexing (Raw fast5 files). Reads were mapped with minimap2 option ‘-ax map-ont’ to the *C. elegans* WS287 transcriptome and pooled into an expression matrix via a dedicated Python script. Differential expression analysis was performed on R with edgeR and limma packages. Genes < 1 counts per million (cpm) were excluded, and then the data were normalized (limma-voom) and transformed into log2, a linear model which is adjusted to the data for each gene. The contrasts were then extracted and compared to the linear model according to the empirical Bayesian method, and the obtained *p*-values were adjusted by the Bonferroni method [37] to obtain the final differential expression results. A volcano plot of all of the genes’ status of differential expression was produced using Glimma using the foldchange as the X-axis and the negative log10 of the adjusted *p*-value as Y-axis. A heatmap of compared relative expression between conditions of 136 genes, 36 differentially expressed genes, and 100 highest foldchange genes (50 up-regulated, 50 down-regulated) not classified as differentially expressed, was produced using ggplots. Enrichment analysis was performed with the R package gprofiler2 [38], which queries public APIs from several databases and determines statistically significant results based on the number of genes impacted and the total number of genes attached for each ontology.

## 3. Results

### 3.1. Expression of the GFP::PCN-1 Fusion Protein Allows Specific Targeting under a Microbeam of Primordial Germ Region (Z2-Z3) Cells for the Monitoring of Germline and Vulva Developments

When undertaking targeted irradiation of specific regions within a multicellular organism, two essential factors become critical: (i) the precise identification of the target; (ii) the accurate immobilization of the target. 

(i) Identifying cells unambiguously is probably the most important and difficult part for a targeted irradiation experiment. Thus, cell identification should be assisted by the availability of promoter fusions with the green fluorescent protein (GFP) and the use of established and well-characterized transgenic strains that allow the identification and monitoring of specific cell types by fluorescence microscopy in live animals. 

Since *C. elegans* is transparent, individual cells can be distinguished by fluorescence microscopy and followed in living worms. At hatching, L1 hermaphrodite larvae have two primordial germ cells (Z2 and Z3) sandwiched between two somatic gonad precursors (Z1 and Z4) and their surrounding basement membrane (Figure 1a). These fours cells remain mitotically quiescent until the mild-L1 stage. Z2 and Z3 require nutritional, and cell-cell signals to proliferate. In order to identify the primordial gonadal region specifically, we selected the GZ264 transgenic strain which expresses the GFP-tagged protein PCN-1 (GFP::PCN-1) under the PIE-1 germline expression promotor [39,40] (Figure 1b). Z2 and Z3 begin proliferating in mid-L1 to populate the gonad with germ cells [41]. This fluorescent reporter is expressed in the primordial germ line Z2 and Z3 and is localized in the cell nucleus in the S-phase. At the early-L1 stage, GFP::PCN-1 is only expressed in the nucleus of Z2 and Z3 as shown in the Figure 1a. The Z2-Z3 area can be unambiguously distinguished by fluorescence microscopy and thus targeted with the microbeam. (Figure 1b). In adult worms, the expression of this fluorescent reporter is also conserved and helps in the characterization of the gonadal system development. Gonads, oocytes, and in utero fertilized embryos can be easily observed by the presence of the GFP in their nuclei (Figure 1a,b). 

(ii) For the immobilization of the intended target, we applied established charged-particle microbeam irradiation techniques, originally designed for in vitro and in vivo, cultures, using our custom support dish, as previously detailed in the works of Muggiolu G. et al. and Torfeh E. et al. [14,16], for micro-irradiation and live imaging of *C. elegans* embryos. The sample holder, as shown in Figure 2, offers a stable long-term environment for microscopic analysis and micro-irradiation experiments. We used a bleaching technique to synchronize *C. elegans* cultures at the first larval stage (L1). After bleaching, the embryos were placed on NGM agar plates without food, enabling hatching while preventing further development. The L1 larvae were then maintained on NGM agar plates without a food source until irradiation time (as illustrated in Figure 2a). We minimized the time between hatching and irradiation to promote healthy conditions and reduce the duration of starvation (<20 h). 30 min before irradiation, L1 populations were placed in a specific mounting medium composed of M9 medium supplemented with Tetramisole hydrochloride and Poloxamer-407. Poloxamer 407 is a non-ionic tri-block copolymer (typically with a hydrophobic central block of polypropylene glycol), which is unique in that it undergoes reverse gelatinization, i.e., it changes from a liquid at 4 °C to a gel when it warms at room temperature (~20 °C). This reverse gelatinization helps to immobilize the worms and to recover them rapidly and easily after irradiation. An aliquot ~2 µL was directly deposited on a sterile 4 μm thick polypropylene and immediately covered with an afresh agar pad to maintain immobilized worms in a thin layer of the medium. This procedure resulted in the reproducible placement of the worms in the focal plane of the objective lens (Figure 2b). To prevent desiccation and contamination, the dish was closed with a glass side coverslip. The elapsed time between mounting and irradiation was kept under 1 h to favor healthy conditions. Live imaging using fluorescence microscopy allowed rapid and simple visualization of synchronized and immobilized L1 worms. The Z2-Z3 cells were identified through the expression of the GFP::PCN-1 protein. We chose to irradiate selectively and specifically Z2-Z3 cells (Figure 2c,d). During the experiment, at least 300 nematodes were micro-irradiated with 300 Gy of protons (3 MeV) in the Z2-Z3 area. The selected dose is in agreement with previous experiments performed on *C. elegans*, with doses ranging from 100 to 400 Gy [17,41,42,43]. 

### 3.2. Confocal Microscopy of Radiation-Induced Alterations of Gonadal and Vulval Development in C. elegans following Selective Irradiation

Following irradiation, the worms were promptly retrieved and placed in favorable growth conditions to favor their development. After 3 days, L1 larvae reached their adult stage, characterized by the presence of in utero embryos. Adult worms were chemically fixed using a paraformaldehyde/acetone treatment, then incubated with Hoechst^33342^ and Phalloidin-AF^594^ to label specific subcellular structures such as the nuclear DNA and the a-actin fibers network, respectively. The GFP reporter, GFP::PCN-1, helped in specifying the germ cell lineage.

Confocal imaging revealed the different characteristics of the anatomic structures of the gonads and vulva (Figure 3a). First, the presence of the in utero fertilized embryos (GFP::PCN-1) and the specific muscular architecture of the typical cross-sectional profile of the vulval muscles (PhalloidinAF^594^) were clearly defined, as shown in control samples in Figure 3b. By contrast, in worm populations subjected to Z2-Z3 cell irradiation, notable alterations in gonadal and vulval development were evident (Figure 3c–f). These changes included the absence of expected organs, agenesis of gonads, absence of embryos, and disrupted GFP::PCN-1 expression. The vulval development was similarly compromised, accompanied by disorganized actin fibers and the absence of residual GFP::PCN-1 expression. Additionally, targeted irradiation caused gonadal agenesis, disrupted cell proliferation, and abnormal vulvar eversion (Figure 3e,f). All the irradiated worms at 300 Gy presented the described radiation-induced developmental changes in the selected and targeted tissues. Quantitative analysis on a large number of worms would be necessary to well characterize the radiation-mediated phenotype. It would be possible to plan the studies with parameters such as pharyngeal pumping, locomotion patterns, or food preferences in *C. elegans* following targeted micro-irradiation.

### 3.3. High-Throughput Cell Measurement of Radiation-Induced Developmental Alterations in C. elegans following Selective and Targeted Irradiation (COPAS)

High-magnification fluorescence microscopy is extremely time-consuming and labor intensive [44]. Moreover, because rapid microscopic examination can be performed on only a limited number of animals, the obtained results can be strongly affected by stochastic variations among individuals. We tested the high-throughput characterization of the radiation-induced alterations with a “complex object parametric analysis and sorter” (COPAS) instrument equipped with a profiler system that analyzes up to 100 animals per second [36,45]. To study the GFP::PCN-1 expression at the level of a large population of animals with a quantitative read-out, we used COPAS, which generates fluorescent emission profiles along the antero-posterior axis of the *C. elegans* body. By analyzing large numbers of animals of all sizes and ages at high throughput, we generated a digitized overview of the GFP reporter expression throughout post-larval development after targeted and selective irradiation of the gonad primordium region. For each population analyzed, fluorescence profiles were acquired for hundreds of nematodes from an initial synchronized-stage culture (L1 larvae). As an example, Appendix A illustrates the development of the worm population GZ264 analyzed with COPAS under normal growth conditions and after global irradiation (300 Gy). For each worm in the population, we then converted the corresponding profile into a color-coded representation of the fluorescence intensity. After orienting these profiles, we assembled them so that the short rows at the bottom represent L1 larvae, whereas the top rows correspond to fully-grown adults (Appendix A). As the distribution of worm length varies dramatically between analyzed populations (Appendix A), we generated images in which each row represents the average of all worms of a given length. If no animal of a given length was found, the corresponding row was skipped. As these normalized images provide an overview of GFP expression in the time throughout post-larval development (the length of the worm being a proxy for its age), we refer to them as “chronograms” (Appendix A) [45]. The tissue-specific signature corresponding to the gonadal system is clearly identifiable on the chronograms (Appendix A). From these chronograms, we were also able to extract the length distribution and the expression intensity of the GFP. As illustrated in Appendix A, irradiated worms revealed stunted growth (worm length) and reduced GFP intensity, indicating the status of the gonadal development.

We conducted COPAS analysis on two distinct populations of worms: one group was selectively micro-irradiated with a dose of 300 Gy, while the other served as the control, remaining non-irradiated. Analysis was performed on 450 control and 514 micro-irradiated individual worms, respectively. The resulting chronograms, as illustrated in Figure 4, reveal significant disparities between these two groups. These differences manifest not only in terms of radiation-induced developmental alterations, as evidenced by variations in worm length (TOF), but also in fluorescence intensity (GFP). As depicted in the chronograms, we observed a substantial subpopulation of worms exhibiting high GFP fluorescence intensity in the controls, along with worms exceeding a size of 200 (Figure 4a,b). In contrast, Figure 4b reveals the growth development delay induced by selective irradiation (worm length). In addition, Figure 4c illustrates that irradiated worms displayed lower fluorescence intensity, correlating with their reduced length. Upon a closer examination of worms with a TOF > 200, we noted a pronounced contrast in average fluorescence intensity between the two populations (Figure 4d). This illustrates the absence of cells expressing the GFP reporter under the tissue-specific promoter pie-1, indicating an abnormal gonadal development (delay or absence). These observations obtained on a large number of animals (>400) agree with our previous observations obtained by confocal microscopy. Our findings highlight the importance of integrating targeted irradiation with high-throughput population analysis, such as COPAS, to assess the impact of irradiation on organogenesis and worm development using large populations. 

### 3.4. Transcriptomic Analysis Revealed Differential Expression of Genes Involved in Metabolic Stress and Innate Immune Response Related to Physical Cuticle Injuring

Studying the cellular expression of a cell and/or animal at a given time following an exposure provides valuable information on its physiological status by identifying potential cellular pathways that are quantitatively impacted in response to the induced stress. Transcriptomic analysis reveals the gene expression profile of samples via sequencing of all the gene transcripts and statistical analysis of differential expression between experimental conditions. As previously observed by COPAS analyses and confocal microscopy, irradiation alters the processes of worm development. It is therefore logical to expect this irradiation-induced stress to result in transcriptomic activity aimed at activating stress cellular pathways’ stress response and stopping or slowing non-essential cellular functions to maximize the chances of survival for the organism. Long recovery times post-irradiation would however introduce a great heterogeneity in terms of developmental stage, cell number, and fate. To minimize this constraint and to study the early biological responses induced after targeted irradiation, we decided to perform our experiments using a short recovery period of 3 h post-irradiation. This time frame situates the study in the early response to radiation-induced stress in worms still sharing the same overall characteristics, as irradiation damage has not yet had time to impact their development. This choice also implies accepting some technical constraints, particularly in the RNA extraction procedure. If we consider that the content of total RNA corresponds to 1% of total mass (3% dry weight), a single L1 larva contains ~800 picograms of total RNA (https://wiki.wormbase.org/index.php/Worm_numbers, accessed on 20 September 2023). Thus, to reach the amount of total RNA required to apply third generation sequencing, we decided to pool several replicates of independent irradiation runs. These pooled samples were prepared for sequencing with the cDNA barcoding kit from Oxford Nanopore Technologies (ONT), which allows the multiplexing of pooled libraries, built from small amounts of total RNA (>50 ng), on the same flow cell. For each input cDNA sample, a unique barcode is incorporated into the library of cDNA molecules prepared for sequencing. The resulting reads can then be assigned to their original library according to the barcode sequence. Two sets of complete irradiation experiments were sequenced with this method simultaneously with three flow cells (R.9.4.1, ONT), using the MK1C instrument, with a data collection period of 24 and 48 h (depending on the number of remaining active nanopores). We obtained 15 to 20 million “reads”, with at least 500,000 reads per barcoded sample. 

The sequencing output of these flow cells was then demultiplexed, base-called, and aligned on the reference *C. elegans* transcriptome to obtain a gene expression matrix of all the samples. Differential expression analysis was performed to compare gene expression between the control and the irradiated samples (at 300 Gy) (Figure 5). This process aims to identify differentially expressed (DE) genes, which are genes with an important foldchange between the two conditions and a sufficient global level of expression, to ensure the statistical relevance of this foldchange. This differentially expressed (DE) gene computation identifies 36 genes for the 300 Gy conditions, with 19 being under-expressed and 17 over-expressed (Figure 5).

This identification of DE genes therefore allows us to confirm the presence of a distinct modification of gene expression at the scale of the entire organism. While this number of DE genes does not correspond to what would be expected from a ‘strong’ transcriptomic response, it should be seen in the context of the experimental conditions used. Since irradiation was limited to a region defined by the Z2-Z3 cells in organisms of around 500 cells, identifying a cellular response at the scale of the organism was not a given. Furthermore, the choice of a 3 h post-irradiation time frame and the dose of 300 Gy could potentially limit our analysis to early response genes and choosing a different delay and dose could result in a more extensive transcriptomic response. It is worth noting, nonetheless, that despite the low number of DE genes, an important number of genes exhibit important fold change (363 up-regulated and 263 down-regulated with a >2 foldchange) without being classified as DE due to their low overall expression (Figure 5b). Most of them exhibit consistent expression among experimental replicates, hinting at robust detection of a core set of impacted genes (Figure 5b). Multiple reasons could explain these genes’ low overall expression: insufficient sequencing material, genes with low endogenous expression levels, the onset of activation of stress pathway genes, etc. In any case, these genes, while not being classifiable as DE at the time, hint at the rich gene expression trends that could be uncovered on further production and analysis of other replicate samples to increase statistical significance.

Despite the relatively small amount of DE, it is striking to note that several genes belonging to the same cellular signaling pathways stand out (Appendix A). The first identified pathway concerns the heat shock and oxidative stress responses. This pathway is illustrated by the over-expression of the hsp gene family (hsp-16.1, hsp-16.2 hsp-16.41, hsp-16.48, and hsp-16.49), with fold changes ranging from ~1.29 to ~2.36. In *C. elegans*, the smHSP16s are relatively well studied [46]. The major hsp16 genes are encoded by hsp16-1/hsp16-48 and hsp-2/hsp16-41 gene pairs. These four genes are very similar in their gene structures and amino acid sequence homology [47]. The gene hsp16-2 is known to be increased by reactive oxygen species and electro-magnetic fields [48,49]. The genes hsp16-1 and hsp16-48 are also increased by oxidative stress [50,51]. These genes are not expressed in a normal condition and only induced by stresses, indicating that these genes may not have other roles beyond stress responses. In the future, we plan to validate these radiation-induced metabolic pathways through specific GFP-tagged reporters under hsp16 and/or hsp4 promoters using COPAS and real-time microscopy.

Second, we identified pathways concerning gene families involved in metabolic stress responses. Many of these genes are potentially associated with a stress response of some sort, including metabolism genes such as a cytochrome P450 (cyp-14A5), several UDG glucuronosyl transferases (ugt-26, ugt-29, and ugt-31), and metal-responsive genes (mtl-1 and nspe-1). These genes have been found to be differentially expressed after UV-C exposure [52]. We found an inhibition of these metabolic pathway genes, reflecting a slowing down of the general metabolic cellular pathways specific to the normal functioning of the worm, which illustrates a cellular stress situation (elo-2, elo-5, elo-6, ugt-26, ugt-29, cyp-14A5, etc.). These genes’ families are involved in detoxification processes of small molecules that occur in response to heat shock, reactive oxygen species, and toxic compounds. 

Third, we identified pathways including genes involved in the tissue specific morphogenesis (epithelia, vulva, and gonad) and the innate immune response. It concerns the NPL gene family (for neuropeptide-like protein with nlp-66), which has been found to be differentially expressed at 300 Gy (fold changes ~1.25). We also found the Scl-2 and K08D8.5 genes were also involved in the cuticle defense response to fungus and innate immune response (fold changes ~1.39 and ~1.25, respectively). This observation is in agreement with previous observations that both needle and laser wounds induced pnlp-29::GFP expression in the *C. elegans* epidermis with a similar time course, detected within 1 h of injury [53]. Indeed, the precise damage of epidermal cells realized using femtosecond laser pulses focused on the apical surface of the epidermis creates small wounds, causing a disruption of the epidermis and cuticle. Our observations suggest that targeted and selective irradiation (restricted to the Z2-Z3 region) mimics epidermis “wounding” mediated by IR and that *C. elegans* can respond transcriptionally to this radiation-induced injury. 

Taken together, these results demonstrate our ability to study the cellular response in selectively micro-irradiated samples via the identification of differentially expressed genes within important cellular pathways of the stress response and allows us to depict a schematic representation of the in vivo radiation-induced response after selective and targeted irradiation (Figure 6).

## 4. Discussion

Our main goal in targeted irradiation is to accurately cause damage in precise cellular areas within specific cellular regions in multicellular organisms. Laser ablation is the preferred method for achieving this goal due to its compatibility with microscopy, enabling simultaneous irradiation and observation. This technique has a long history of use in studying *C. elegans* cell lineage and development. Targeted irradiation using ionizing radiation in multicellular organisms faces limitations, including limited access to microbeam facilities and constraints on the number of irradiated animals due to technical challenges associated with sample manipulation and irradiation procedures. Using a charged-particle microbeam requires separate equipment for observation and irradiation, leading to increased complexity involving precise beam–microscope alignment, specialized sample preparation, and synchronization of time-lapse imaging. One significant limitation lies in the real-time monitoring of radiation-induced damage through time-lapse imaging, which generally restricts the study to a small number of individuals and is constrained by beam time availability. Nevertheless, charged-particle microbeams offer a unique advantage by enabling precise control over irradiation dosage down to a single particle, playing a crucial role in studying radiation-induced effects in biological models.

When conducting targeted irradiation within a multicellular organism, two key factors come into play: the precise identification of the target and the immobilization of the organism. We validated a methodology by focusing on the organogenesis of the vulva and gonad, two well-defined and extensively studied developmental systems in *C. elegans*. Our approach involved the precise manipulation of worm populations and targeted irradiation of stem progenitor gonad cells with three MeV protons during a specific developmental stage (L1). This required immobilization, identification, imaging, irradiation, and subsequent recovery of a synchronized population with minimal time and manipulation. To achieve this, we used a specialized mounting medium composed of M9 medium supplemented with Poloxamer-407, a unique non-ionic tri-block copolymer known for its property of reverse gelatinization. This property was instrumental in immobilizing the worms and facilitating their rapid recovery after irradiation. In this study, we chose to employ a high dose of irradiation as a proof of concept. Moving forward, we now have the flexibility to design new experiments with varying doses to explore the dose–response relationship and potentially identify threshold levels. The impact of micro-irradiation on the lifespan and different phenotypic traits could be studied in the future.

Additionally, we employed a *C. elegans* strain expressing the PCN-1::GFP-tagged reporter under the control of the pie-1 promoter. This greatly aided in the identification of gonadal progenitor stem cells through fluorescence microscopy. *C. elegans* is a unique multicellular model for conducting in vivo gene expression studies, thanks to its transparent body, highly predictable cell lineage, and numerous collections of GFP-tagged reporters, making it ideal for precise analyses of radiation-induced effects throughout development. Following irradiation and subsequent animal development, we conducted high-magnification fluorescence microscopy to assess gonadal and vulval development. While this analysis was time-consuming and labor-intensive, it provided valuable insights into the targeted organ development. Specifically, for targeted irradiation at 300 Gy, we observed not only organ absence (agenesis) but also anomalies such as vulvar eversion characterized by cells exhibiting tumor-like uncontrolled development, suggesting that targeted irradiation of the stem cell region induces alterations in tissue-specific cell signaling and cell development. Similar to other studies, we have identified the vulva as a somatic tissue exhibiting radiosensitivity, perhaps because the vulva is one of the few organs dividing post-embryonically [42,43]. 

To comprehensively analyze gonadal development post-irradiation at 300 Gy, we used COPAS to quantitatively evaluate developmental abnormalities in several hundred animals. GFP fluorescence measurements across a significant number of samples reaffirmed our initial observations made through confocal microscopy, confirming the absence of gonads in selective and targeted irradiated populations. Since GFP expression is specific to gonadal cells, we were able to measure GFP fluorescence across a significant number of samples. This approach provided us with a complementary dataset that reaffirmed the initial observations made through confocal microscopy, namely, the decrease in fluorescence, and consequently, the radiation-induced modifications in gonad development in irradiated populations.

Finally, we harnessed the latest advanced transcriptome analysis technologies, demonstrating the feasibility of conducting such experiments on a relatively small number of individuals. We identified a radiation-induced response through the expression of genes involved in pathways such as heat shock and oxidative stress responses (over-expression of the *hsp* gene family, *hsp-16.1*, *hsp-16.2 hsp-16.41*, *hsp-16.48,* and *hsp-16.49*). Moreover, our findings revealed a response mediated by the cuticle defense response (*nlp-66*) and by the innate immune response (*Scl-2*), highlighting the significance of studying at the organism level, particularly in extensively characterized organisms like *C. elegans*. Similar phenomena have been observed in instances of biomechanical damage caused by factors such as bacterial infections, mechanical abrasions, or laser injury. It would be interesting to test the *C. elegans* strain expressing the inducible *pnlp-29*::GFP reporter to demonstrate that *C. elegans* responds to physical injury of the epidermis mediated by ionizing radiation. It has been shown that the activation of *nlp-29* via wounding suggests that in *C. elegans*, tissue damage triggers an innate immune response [53]. It would be interesting to test this hypothesis using targeted irradiation at different doses of exposure and at different developmental stages. Exploring the cuticle and epidermis of *C. elegans* as a model for understanding radiation-induced responses in epithelial tissues holds significant promise. The data presented in our study open exciting opportunities to investigate the cellular and molecular mechanisms underlying radiodermatitis. Although the *C. elegans* cuticle constitutes a highly specialized extracellular matrix (ECM) with intricate and unique features, its biogenesis shares fundamental molecules, mechanisms, and pathways with vertebrates. This commonality makes *C. elegans* an invaluable experimental system for dissecting ECM formation. This research allows for the examination of alterations in gene expression, signaling pathways, and cellular responses within these tissues, leading to the identification of crucial molecular components and mechanisms (for example, the MAP kinase signaling pathways). These findings offer valuable insights into the molecular and cellular basis mechanism of radiodermatitis [54,55] in response to radiation exposure and hold significant promise for advancing future prospects in radiation therapy.

## 5. Conclusions

In conclusion, the present works offers a comprehensive approach to investigate targeted irradiation in multicellular organisms, specifically focusing on gonadal and vulval development in *C. elegans*. We have designed an efficient strategy for targeted irradiation of specific stem cell regions and analyzed many individuals, yielding valuable insights into the impact of radiation on organ development. By combining fluorescence microscopy, COPAS-based quantitative measurements, and transcriptome analysis, we have revealed molecular mechanisms underlying radiation-induced responses. These mechanisms encompass the expected involvement of heat shock proteins and reactive oxygen species, as well as an unforeseen pathway related to cuticle defense and the innate immune response. This unexpected finding opens exciting new research perspectives within the context of radiodermatitis. Overall, our work contributes to the comprehension of how radiation exposure influences gene expression, signaling pathways, and cellular reactions in epithelial tissues, ultimately advancing the understanding of radiodermatitis induction and promising avenues for future advancements in radiation therapy.

## Figures and Tables

**Figure 1 biology-12-01372-f001:**
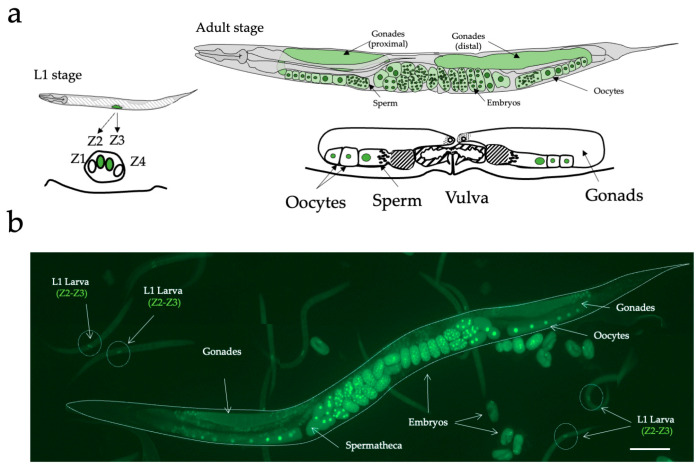
GZ264 transgenic *C. elegans* strain. (**a**) Schematic representation of post-embryonic hermaphrodite gonadal and vulval development in wild-type hermaphrodites. Comparative size of gonads at different stages is not to scale. Each diagram represents two stages of post-embryonic development of the gonad and the vulva. L1 stage: Two cells, Z1 and Z4, are the progenitors of the somatic tissues of gonad. Two other cells, Z2 and Z3, give rise to the germ line. The line below the gonad represents the underlying ventral hypodermis. Adult stage: somatic gonadal development is completed. The vulval invagination has everted to form the mature lips of the vulva. Oocytes and sperm are visible in the proximal part of the gonad. Description and scheme according to Seydoux et al., 1993 [30]. (**b**) Fluorescence imaging of different developmental stages expressing the GFP::PCN-1: adult, embryos, and L1 larvae. Scale bar: 100 µm.

**Figure 2 biology-12-01372-f002:**
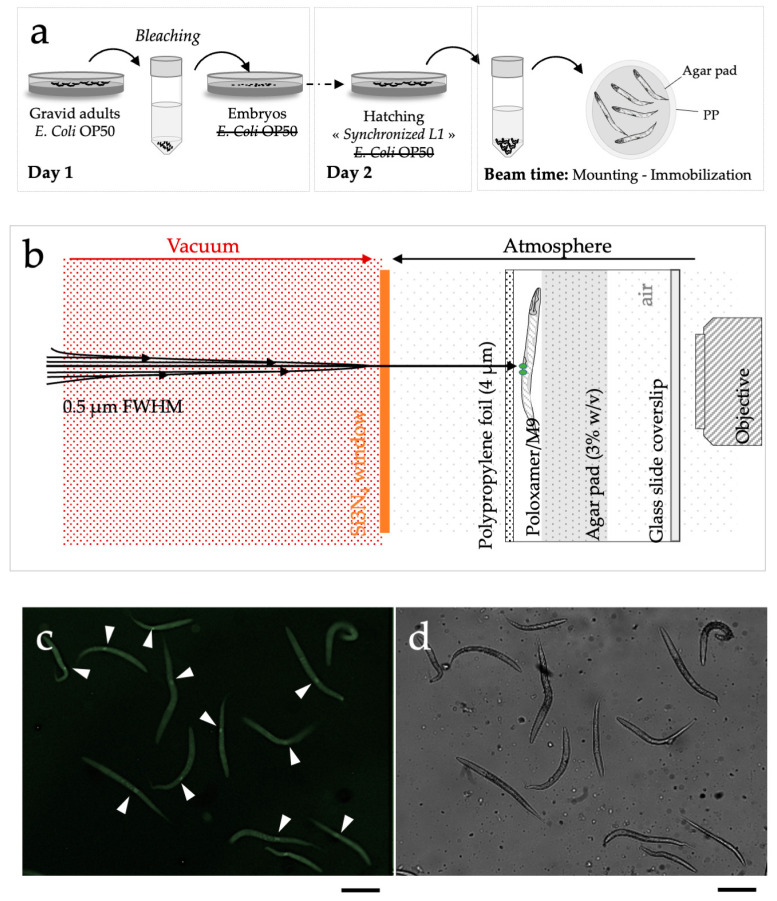
Schematic representation of the different steps needed for micro-irradiation of *C. elegans* larvae (early L1 stage). (**a**) Preparation of large populations of early *C. elegans* L1 larvae by bleaching in the absence of *E. coli* OP50. (**b**) Detailed scheme of the microbeam end- station. 30 min before irradiation, an aliquot ~2 µL was directly deposited on a sterile 4 μm thick polypropylene (PP) foil and immediately covered with an afresh agar pad to maintain immobilized worms in a thin layer of medium. To prevent desiccation and contamination, the dish was closed with a glass side coverslip. Z2-Z3 nuclei were targeted using online fluorescence microscopy. The beam was positioned on the targeted cell. (**c**) PCN1::GFP detection in Z2-Z3 cells in synchronized early L1 using on-line fluorescence microscopy as obtained in real experimental conditions. White arrows indicate Z2-Z3 GFP-positive cells in synchronized early L1 larvae. (**d**) Synchronized early L1 larvae visualized using phase contrast imaging. scale bar: 100 µm.

**Figure 3 biology-12-01372-f003:**
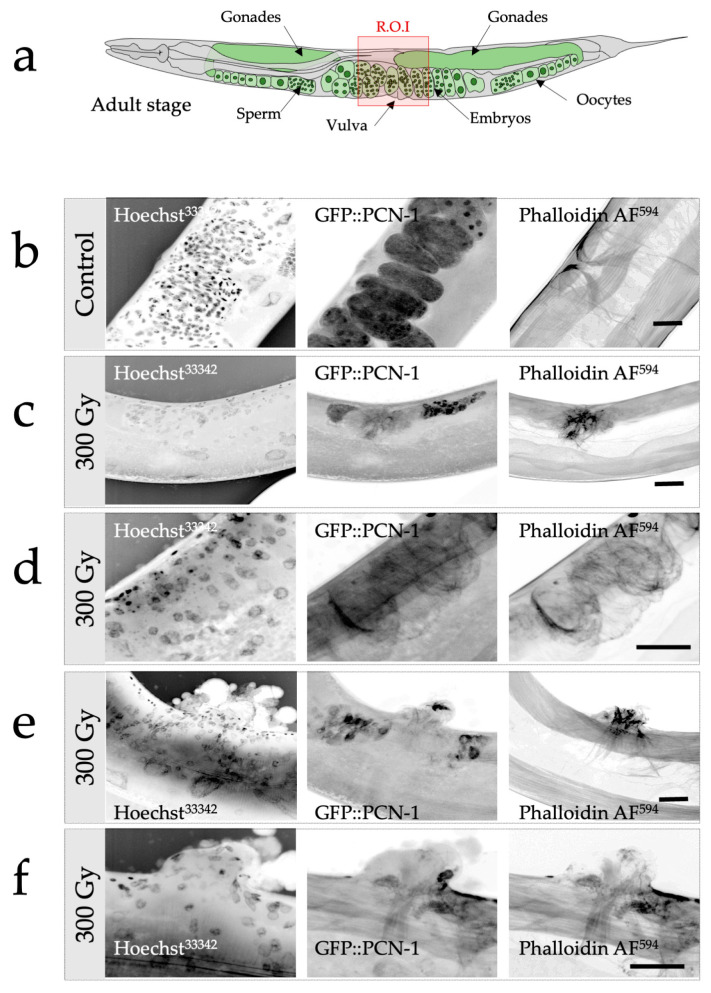
Radiation-induced alterations of gonadal and vulval development in *C. elegans* following selective and targeted irradiation analyzed by confocal microscopy. (**a**) Schematic representation of the gonads and vulva in an adult worm (control). The region of interest (ROI) is depicted with the red square. (**b**) Confocal imaging of the ROI of an adult worm. Nuclei (DNA) are visualized using Hoechst^33342^ (left column), in utero fertilized embryos are revealed thanks to the GFP reporter (middle column), and the vulvar muscles with their typical cross-sectional profile are detected with the help of phalloidin (actin fibers; right column). (**c**–**f**). Confocal imaging of the ROI of structural alterations detected in irradiated worms (300 Gy). Several configurations were observed. (**c**) Gonadal and vulval agenesis, absence of gonadal and vulval development with a limited actin fibrillar network, no embryos detected, absence of the typical cross-sectional profile of the vulval muscles. (**d**) Gonadal agenesis and vulvar developmental anomalies. Evidence of tissue disorganization illustrated by the actin fibrillar network and absence of gonads. (**e**,**f**) Gonadal agenesis and abnormal vulval eversion. Tissular disorganization, cell nuclei proliferation and disorganization, alteration of the actin fibrillar network, and evidence of abnormal vulvar eversion and agenesis. Scale bar: 10 µm.

**Figure 4 biology-12-01372-f004:**
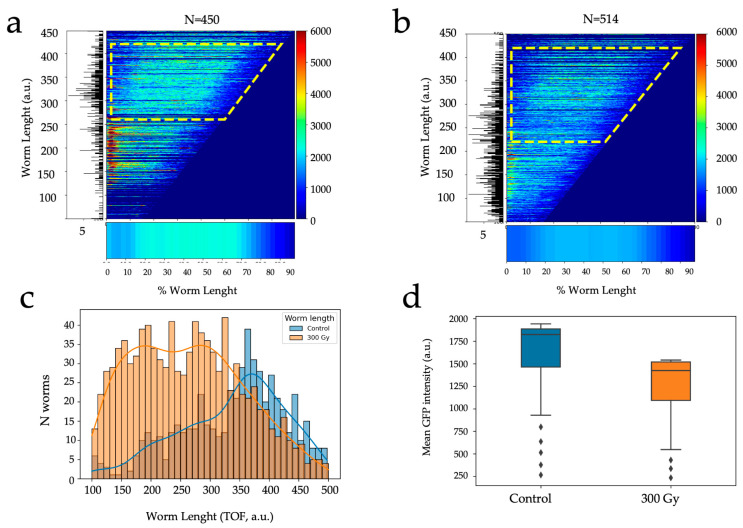
Radiation-induced developmental alterations in *C. elegans* following selective irradiation, analyzed by COPAS. (**a**) Longitudinal fluorescence profiles (GFP) in control population as a function of development (worm length, TOF). The appearance of gonads at the L3/L4 stage and above (TOF > 250–300, dotted yellow region of interest) is converted into a bar where the calculated mean GFP longitudinal intensity profile is indicated by the length corresponding to the animal’s size and the color codes for the fluorescence intensity (bottom figure). The corresponding distribution of worm length is shown on the left panel. (**b**) Longitudinal profiles of GFP fluorescence in the population micro-irradiated at 300 Gy. The appearance of gonads at the L3/L4 stage and above (TOF > 250–300, dotted yellow region of interest) is visible in the calculated mean GFP longitudinal profile (bottom figure). (**c**) Length distribution of worms in control population (blue) and irradiated population (orange). (**d**) Average fluorescence intensity in worms above L3/L4 stage (control, N = 450; 300 Gy, N = 514).

**Figure 5 biology-12-01372-f005:**
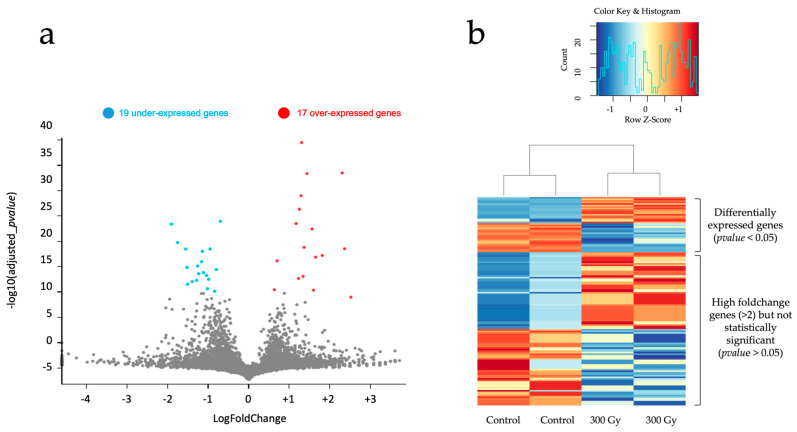
Bulk transcriptomic analysis of cellular expression differences in two *C. elegans* samples containing multiple pooled replicates following selective irradiation. (**a**) Volcano plot of differentially expressed (DE, gray dots) genes in the 300 Gy condition compared to the control. DE genes were computed by fitting a linear model on each gene, extracting contrasts and performing an empirical Bayes test. The genes with a *p*-value < 0.05 (after Bonferroni correction) are considered as DE. (**b**) Heatmap of relative gene expression between control and 300 Gy samples on the 36 DE genes and 100 high foldchange genes (>2), 50 up-regulated and 50 down-regulated genes, not classified as DE. The 100 high foldchange genes selected were the top 50 highest foldchange in up-regulation and down-regulation.

**Figure 6 biology-12-01372-f006:**
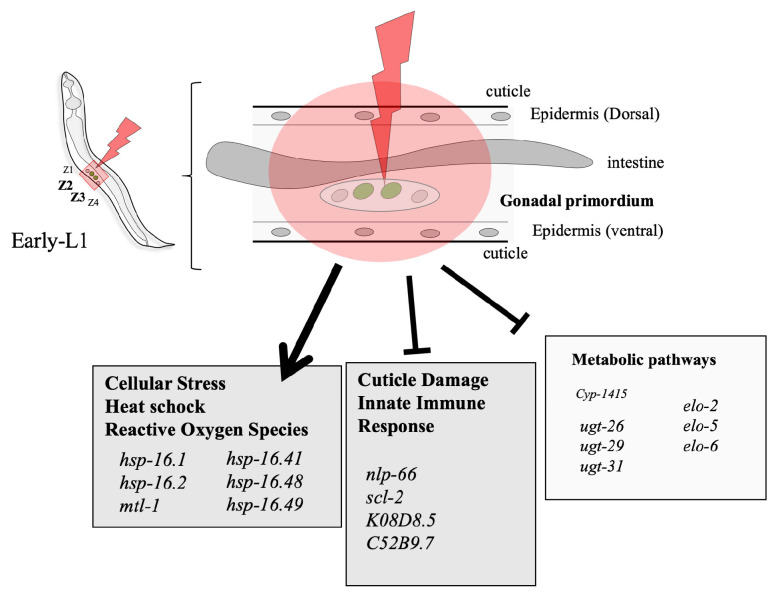
Schematic representation of data obtained from differential expression analysis on *C. elegans* bulk transcriptomes, following selective irradiation performed on gonadal primordium cells at early L1 stage. Genes belonging to distinct pathways of interest were identified as differentially expressed 3 h post-irradiation.

## Data Availability

The raw data supporting the conclusions of this article will be made available by the authors without undue reservation.

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
