# Peer review of "Proton Microbeam Targeted Irradiation of the Gonad Primordium Region Induces Developmental Alterations Associated with Heat Shock Responses and Cuticle Defense in Caenorhabditis elegans"

_biology, 2023, doi:10.3390/biology12111372_

Round 1

Reviewer 1 Report

Comments and Suggestions for Authors

Author Response

Dear Reviewer,

Thank you for comments and your time and efforts to review our manuscript.

We have made changes in line with the comments and have taken as much of the feedback into consideration as possible.

Thank you for your time and consideration.

Sincerely,

H. Seznec

Reviewer 2 Report

Comments and Suggestions for Authors

The work presented is new, interesting and well constructed.

The authors consider the results of changes in gene expression patterns, the response of the worm's immune system to radiation, and an increase in the level of ROS and DNA damage as an opportunity to understand the development of radiodermatitis in humans. Why do the authors so easily transfer data on the metabolism of worms to humans? Are there orthologous genes in this species and is the repair system, radiosensitivity, etc. different?

What do the authors think about the possible bystander effect under these experimental irradiation conditions?

Author Response

Dear Reviewer,

Thank you for comments and your time and efforts to review our manuscript.

We have made changes in line with the comments and have taken as much of the feedback into consideration as possible.

Thank you for your time and consideration.

Sincerely,

Hervé Seznec
